# Hyperspectral Identification of Chlorophyll Fluorescence Parameters of *Suaeda salsa* in Coastal Wetlands

**Wei Zheng** [1,2,3,4], **Xia Lu** [1,2,3,4,*], **Yu Li** [1,2,3,4], **Shan Li** [1,2,3,4] **and Yuanzhi Zhang** [5,6]

1   Jiangsu Key Laboratory of Marine Bioresources and Environment, Jiangsu Ocean University, Lianyungang 222005, China; zwchengdu@jou.edu.cn (W.Z.); 2006000032@jou.edu.cn (Y.L.); lshenan@jou.edu.cn (S.L.)
2   Jiangsu Key Laboratory of Marine Biotechnology, Jiangsu Ocean University, Lianyungang 222005, China
3   Co-Innovation Center of Jiangsu Marine Bio-Industry Technology, Jiangsu Ocean University, Lianyungang 222005, China
4   Faculty of Ocean Technology and Surveying and Mapping, Jiangsu Ocean University, Lianyungang 222005, China
5   Center for Housing Innovations and Institute of Asia-Pacific Studies, Faculty of Social Science, Chinese University of Hong Kong, Hong Kong, China; yuanzhizhang@cuhk.edu.hk
6   Department of Marine Technologies, School of Marine Science, Nanjing University of Information Science and Technology, Nanjing 210044, China
*   Correspondence: 2008000070@jou.edu.cn; Tel.: +86-188-8885-3470

**Abstract:** The stomata of *Suaeda salsa* are closed and the photosynthetic efficiency is decreased under conditions of water–salt imbalance, with the change to photosynthesis closely related to the chlorophyll fluorescence parameters of the photosystem PSII. Accordingly, chlorophyll fluorescence parameters were selected to monitor the growth status of *Suaeda salsa* in coastal wetlands under conditions of water and salt. Taking *Suaeda salsa* in coastal wetlands as the research object, we set up five groundwater levels (0 cm, −5 cm, −10 cm, −20 cm, and −30 cm) and six NaCl salt concentrations (0%, 0.5%, 1%, 1.5%, 2%, and 2.5%) to carry out independent tests of *Suaeda salsa* potted plants and measured the canopy reflectance spectrum and chlorophyll fluorescence parameters of *Suaeda salsa*. A polynomial regression method was used to carry out hyperspectral identification of *Suaeda salsa* chlorophyll fluorescence parameters under water and salt stress. The results indicated that the chlorophyll fluorescence parameters $Fv/Fm$, $Fm'$, and $\Phi PSII$ of *Suaeda salsa* showed significant relationships with vegetation index under water and salt conditions. The sensitive canopy band ranges of *Suaeda salsa* under water and salt conditions were 680–750 nm, 480–560 nm, 950–1000 nm, 1800–1850 nm, and 1890–1910 nm. Based on the spectrum and the first-order differential spectrum, the spectral ratio of A/B was constructed to analyze the correlation between it and the chlorophyll fluorescence parameters of *Suaeda salsa*. We constructed thirteen new vegetation indices. In addition, we discovered that the hyperspectral vegetation index $D_{690}/D_{1320}$ retrieved *Suaeda* chlorophyll fluorescence parameter $Fv/Fm$ with the highest accuracy, with a multiple determination coefficient $R^2$ of 0.813 and an $RMSE$ of 0.042, and that $D_{725}/D_{1284}$ retrieved *Suaeda* chlorophyll fluorescence parameter $\Phi PSII$ model with the highest accuracy, with a multiple determination coefficient $R^2$ of 0.848 and an $RMSE$ of 0.096. The hyperspectral vegetation index can be used to retrieve the chlorophyll fluorescence parameters of *Suaeda salsa* in coastal wetlands under water and salt conditions, providing theoretical and technical support for future large-scale remote sensing inversion of chlorophyll fluorescence parameters.

**Keywords:** coastal wetland; *Suaeda salsa*; water and salt conditions; hyperspectral; chlorophyll fluorescence parameters

## 1. Introduction

Coastal wetlands are located at the junction of land-sea ecosystems [1]. As a result, fluctuation in salinity is one of the key features in wetlands, animals, and plants under

prolonged stress with a variation of salinities [2,3]. In coastal wetlands of Jiangsu Province, the salt marsh plant *Suaeda* is common, while the dominant halophyte *Suaeda* can absorb salt and heavy metals in soil, increase soil fertility, and improve the ecological environment of coastal wetlands [4,5]. In recent years, seawater erosion, coastal reclamation, and invasion of alien species have changed the groundwater level and salinity of coastal wetlands, causing the growth of *Suaeda salsa* under dual stresses of water and salinity [6].

The growth and development of plants cannot be separated from water and salt. However, when water and salt contents are too high or too low, plant growth is restricted [7]. The plant growth response to water and salt is very complex. With a synergistic reaction above and below ground, the influence of an arid environment on the aboveground of plant parts is mainly related to stomatal closure and a reduction of $CO_2$, which leads to the reduction of photosynthetic efficiency [8]. Changes to photosynthesis are directly related to the chlorophyll fluorescence parameters of the PSII photosystem [9]. Photosynthesis of plants consists of a light reaction and a dark reaction. Analysis of chlorophyll fluorescence parameters can quickly determine the photochemical efficiency of plant photosystem PSII without damage, representing an important method of diagnosing the operation of plant photosystem PSII and analyzing the mechanism of plant response to stress [10]. Accordingly, monitoring different chlorophyll fluorescence parameters can provide critical information about how plant photosynthesis dynamics change in the face of multiple environmental stresses [11].

Chlorophyll fluorescence is the endogenous light emitted by a plant itself, and it participates in energy distribution in the plant together with photosynthesis. Photosynthesis is highly sensitive to responses involving environmental factors. Different physiological factors and environmental conditions cause differences in plant photosynthetic characteristics. With the help of plant reflectance spectra and chlorophyll fluorescence information, changes to plant physiological characteristics can be monitored over time [12].

In recent years, with the development of hyperspectral remote sensing technology, the spectral information of plants has gradually become a research hotspot [13]. Plant reflectance spectra can reflect the nutritional status of a plant and changes in its environment, allowing its growth status to be monitored [14]. Lichtenthaler et al. [15,16] found that chlorophyll fluorescence was emitted from spectra ranging from 600 to 800 nm, and had two distinct peaks, between 685 and 690 nm and 730 and 740 nm. In addition, the largest peaks were located at 690 nm and 740 nm, respectively. Zarco-Tejada et al. [17,18] found that the hyperspectral vegetation indices of $R_{680}/R_{630}$, $R_{685}/R_{630}$, $R_{687}/R_{630}$ and $R_{690}/R_{630}$ (R stands for reflectance) were sensitive to the fluorescence parameter *Fv/Fm*. Tan et al. [19] used hyperspectral vegetation indices to monitor the chlorophyll fluorescence parameters of corn *Fv/Fm*. To estimate the growth of rice, Zhang et al. [20] used spectral reflectance to estimate chlorophyll fluorescence parameters and suggested that chlorophyll fluorescence parameters can monitor the oxygen stress effect on rice roots. These studies have demonstrated the feasibility and importance of hyperspectral remote sensing in monitoring chlorophyll fluorescence parameters.

Many studies have undertaken hyperspectral monitoring of the chlorophyll fluorescence parameters of green bamboo leaves [21], wheat leaves [22], corn leaves [19], cotton leaves [23,24], and cowpea leaves [25]. Relatively few, however, have undertaken hyperspectral identification of the chlorophyll fluorescence parameters of coastal wetland plants such as Spartina, *Suaeda* [26], and *Phragmites australis*.

Taking a new perspective of combining spectral characteristics with plant chlorophyll fluorescence parameters during the interaction of water and salt stresses, this study investigated the hyperspectral identification of chlorophyll fluorescence parameters of *Suaeda salsa*. Its objectives were to (1) monitor the chlorophyll fluorescence parameters of *Suaeda salsa* susceptible to water and salt stresses, (2) retrieve the hyperspectral vegetation indices sensitive to the chlorophyll fluorescence parameters of *Suaeda salsa*, and (3) provide a reference for estimating the photosynthetic and physiological parameters of *Suaeda salsa* under multiple stresses in degraded coastal wetlands.

## 2. Study Area and Methods

### 2.1. Study Area

The Linhong Estuary Wetland is located in northeast Haizhou District, Lianyungang City (119°09′53.0″~119°16′13.8″E, 34°39′32.0″~34°48′52.8″N), Jiangsu Province, China (Figure 1) and has an area of about 2353.10 ha. The tide in the area is irregular half-diurnal, with an average tidal range of 3.68 m. Suaeda is covered by water in high tides. The study area is located in the transitional zone between the warm temperate zone and the subtropical zone, with an annual average temperature of 14 °C. The average temperature in summer is 14–25 °C, while the average temperature in winter is 5–14 °C The salinity range of groundwater is 0.2–16%, and the average salinity is about 3%. There are some changes in salinity in summer and winter, but very small.

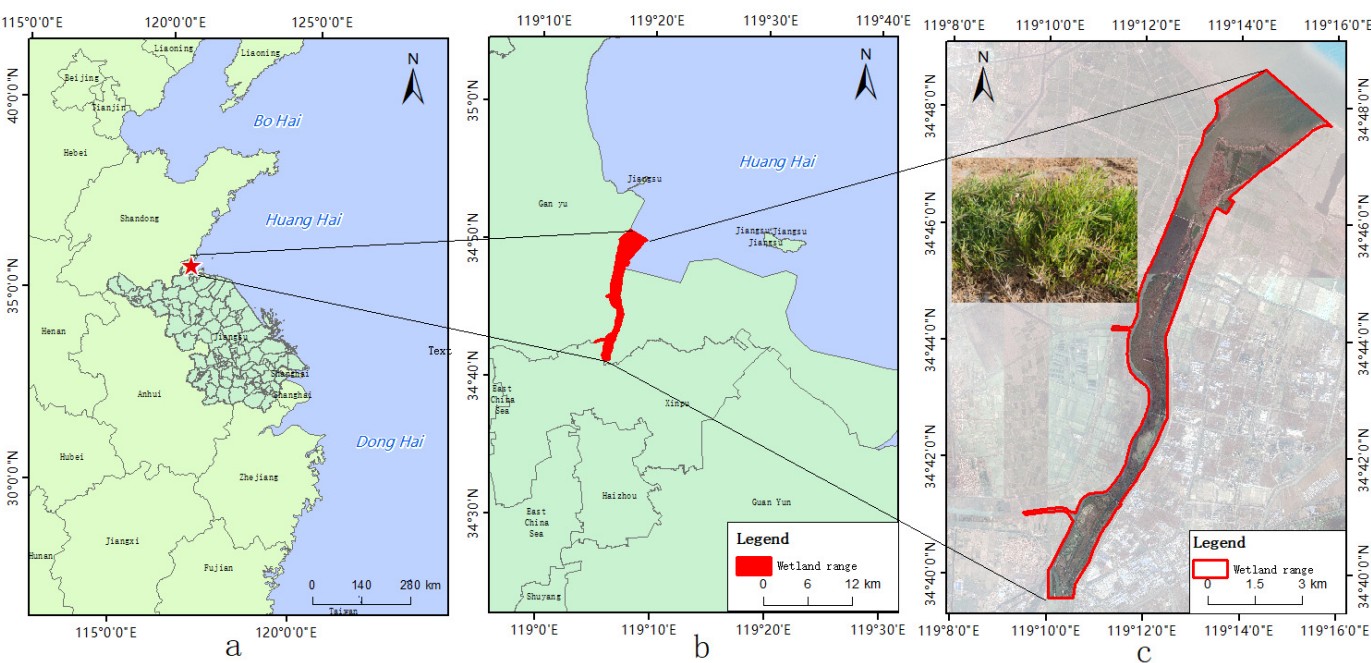

**Figure 1.** Location of the study area. ((**a**) represents the location of the study area in China, (**b**) represents the location of the study area in Lianyungang City, (**c**) represents the specific location of the study area).

It is home to many plant species including 9 families, 10 species of woody plants, and 79 herbaceous plants. The dominant plant communities include reeds, Spartina alterniflora, and *Suaeda salsa*. The natural coastline of the Linhong Estuary Wetland is up to 4.1 km, with a natural tidal flat wetland running near the wetland into the sea. It is an important water bird habitat.

### 2.2. Experimental Design

#### 2.2.1. Field Survey and Experiment Design

*Suaeda salsa* seedlings of similar heights (13–15 cm ± 0.5 cm for plant height, 5–8 cm ± 0.5 cm for root length) were collected from the same experimental site in the Linhong Estuary Wetland on 12 May 2019. Seedlings were planted in plastic pots of different heights (10–40 cm) but the same diameter (20 cm), containing dry fine sands of 20–40 mesh. Pot heights were 10 cm, 15 cm, 20 cm, 30 cm, and 40 cm, respectively.

The treatments combined five levels of water table with six salt levels in an independent design (water table × salt). The pots were placed in sinks with 10 cm depth to control the water table of 0 cm, −5 cm, −10 cm, −20 cm, and −30 cm (Figure 2). Six salt levels of 0–2.5% NaCl were designed in different sets of sinks.

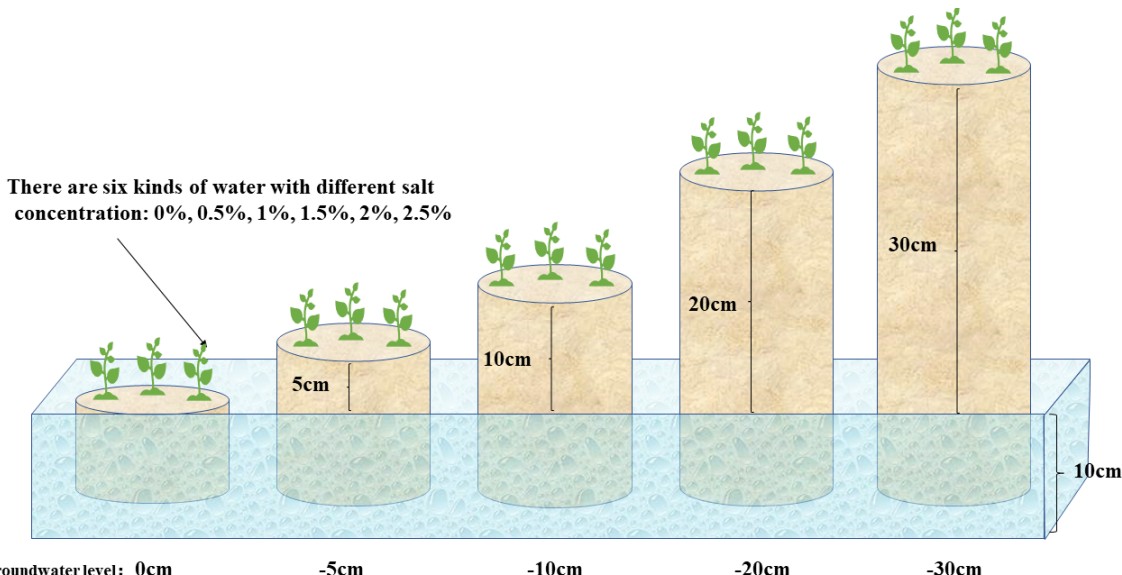

There are six kinds of water with different salt concentration: 0%, 0.5%, 1%, 1.5%, 2%, 2.5%

5cm

10cm

20cm

30cm

10cm

Groundwater level: 0cm    -5cm    -10cm    -20cm    -30cm

**Figure 2.** Schematic diagram of pot experiment scheme design.

Watering of the *Suaeda salsa* seedlings began with different levels of NaCl solutions after a 2-week incubation. About a week later, when the salt concentrations reached the desired treatment levels, the pots were placed into sinks filled with different salt solutions. The salt solutions in the sinks were replaced every third day to avoid significant changes to salt concentration. The pots had drainage holes in the bottom, covered by mesh to prevent sand seepage while allowing water ready entry into the pot. Ninety pots were divided into six sets by salt gradient treatment, with each set containing three replicates.

### 2.2.2. Measurement of Chlorophyll Fluorescence Parameters of *Suaeda salsa*

Based on the principle of difference methods, the portable photosynthesis measurement system (Licor-6400 XT-40, US, equipped with red and blue light sources and $CO_2$ injection system, see Figure 3) was used to measure chlorophyll fluorescence parameters of *Suaeda salsa* (*Fo*, Fm, *Fv/Fm*, *Fo′*, *Fm′*, *Fv′/Fm′*, *ΦPSII*, *qP*, *NPQ*, *ETR*). Among them, *Fo* and *Fm* represent the minimum initial and maximum fluorescence under dark responses, respectively. *Fv/Fm* is the *PSII* original light energy conversion efficiency. *Fo′* and *Fm′* denotes the minimum and maximum fluorescence under light responses, respectively. *Fv′/Fm′* is excitation energy capture efficiency of the *PSII* reaction center opened under light. *ΦPSII* represents the actual *PSII* photochemical quantum efficiency of in the presence of acting light. *qP* represents photochemical quenching coefficient, *NPQ* is non-photochemical quenching coefficient, and *ETR* refers to apparent electron transfer efficiency [10].

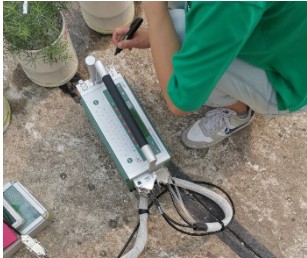

**Figure 3.** Determination of chlorophyll fluorescence parameters.

We selected the sunny weather between 8:30 and 11:30 a.m., using constant photosynthetically active radiation (PAR; 1000 µmol photons m$^{-2}$ s$^{-1}$), the atmospheric concentration of $CO_2$, ambient temperature, and humidity, employing an IRGA. The net $CO_2$

assimilation rate (A), stomatal conductance ($g_s$), and transpiration rate (E) were evaluated. Water use efficiency (WUE) was calculated by the A/E ratio. The evaluations were carried out in the last three days of exposure to the treatments, with five leaves per plant being analyzed [27]. The specific measurement process is shown in the Figure 3.

### 2.2.3. Measurements of the *Suaeda salsa* Canopy Reflectance Spectra

On the 46th day of the experiment, all of the canopy reflectance spectra of *Suaeda salsa* were measured by using an SVC (Spectra Vista Corporation, Poughkeepsie, NY, USA) HR-1024i field-portable spectroradiometer (SVC Inc., USA). The spectrometer covers ultraviolet, visible and near-infrared wavelengths from 350 nm to 2500 nm. In the range of 350–1000 nm, 1000–1850 nm, and 1850–2500 nm, the high spectral resolution is less than 3.5 nm, 9.5 nm, and 6.5nm, respectively. We measured *Suaeda salsa* canopy reflectance spectra from 100 cm above the canopy using 25° field-of-view fiber optics between 10 a.m. and 2 p.m. (Beijing time). The spectral measurement scan time for each pot was 5 s, with the average reflectance spectra of 10 measurements taken as the final spectrum of each pot.

Hyperspectral reflectance spectra of *Suaeda salsa* at the canopy level (see Figure 4) should be preprocessed in order to the further research. First, the abnormal hyperspectral reflectance due to light or wind conditions were deleted. Second, the spectral data were resampled by using the SVC HR-1024i software. Finally, the Savitzky–Golay (S–G) smoothing filter was applied to smooth the reflectance data in order to reduce the instrument and environment noise.

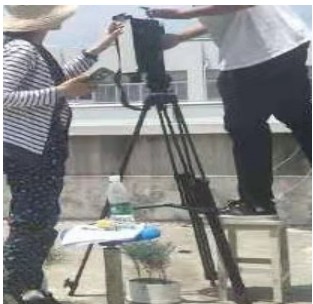

**Figure 4.** Measurement of reflectance spectrum.

### 2.3. Analytical Method

Abnormal hyperspectral data were first deleted and resampled by setting the sampling interval to 1 nm using the manufacturer's program of SVC HR-1024i. Reflectance data were first smoothed with a 5-point moving average to suppress instrumental and environmental noise.

The correlation relationship between chlorophyll fluorescence parameters and the reflectance spectra and first-order differential spectra ratio of the *Suaeda salsa* canopy vegetation indices was analyzed by performing SG filtering and first-order differential processing based on the MatlabR2017b (MathWorks, Natick, MA, USA) programming software.

Two-factor analysis of variance and correlation analysis between hyperspectral data and chlorophyll fluorescence (CF) were carried out in the Statistical Package for the Social Sciences (SPSS 22.0). The relative hyperspectral indices sensitive to CF were selected and the precision of identification models evaluated by a multiple determination coefficient ($R^2$) and a root mean square error (*RMSE*). A total of 90 pots of *Suaeda salsa* were used in the experiment, 60 of which were randomly selected as the experimental group for model construction, with the remaining 30 used as the verification group.

The error scatter plots between the vegetation indices and the chlorophyll fluorescence parameters were drawn in the originpro2020 software (OriginLab, Northampton, MA, USA).

## 3. Results and Analysis

*3.1. Response Analysis of Chlorophyll Fluorescence Parameters of Suaeda salsa to Water and Salt Stress*

Figure 5 shows changes in *Suaeda salsa* chlorophyll fluorescence parameters under different groundwater levels and salt concentrations.

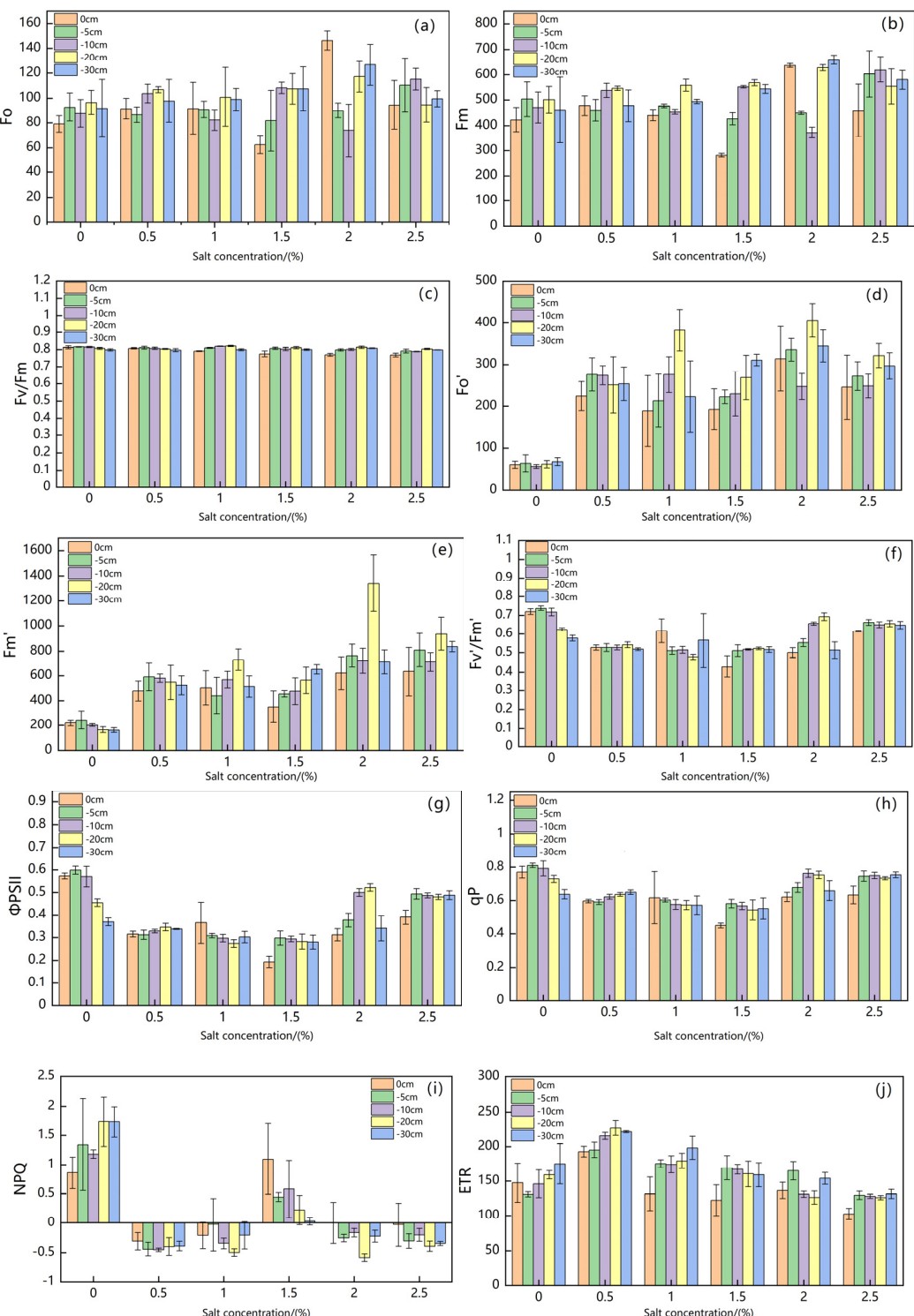

**Figure 5.** Changes of Suaeda salsa Chlorophyll Fluorescence Parameters ($Fo$ (**a**), $Fm$ (**b**), $Fv/Fm$ (**c**), $Fo'$ (**d**), $Fm'$ (**e**), $Fv'/Fm'$ (**f**), $\Phi PSII$ (**g**), $qP$ (**h**), $NPQ$ (**i**), $ETR$ (**j**)) under Different Groundwater Levels and Different Salt Concentrations.

Table 1 shows the results of an analysis of the response of chlorophyll fluorescence parameters of *Suaeda salsa* in coastal wetlands to water and salt stress. *Fo* showed no significant relationships with salt concentration and groundwater level, but exhibited a significant relationship with water–salt interaction stress ($p < 0.01$). *Fm* showed no significant relationship with salt concentration, but exhibited a significant relationship with groundwater level and water–salt interaction stress ($p < 0.05$). $\frac{Fv}{Fm}$, *Fm′*, and $\Phi PSII$ showed extremely significant relationships with salt concentration, groundwater level, and water–salt interaction stress ($p < 0.001$). *Fo′* and *ETR* showed extremely significant relationships with salt concentration and groundwater level ($p < 0.001$) and also exhibited significant relationships with water–salt interaction stress ($p < 0.01$). *Fv′/Fm′* showed an extremely significant relationship with salt concentration and water–salt interaction stress ($p < 0.001$), but exhibited no significant relationship with groundwater level. *qP* showed extremely significant relationships with salt concentration and water–salt interaction stress ($p < 0.001$) and also exhibited significant relationship with groundwater level ($p < 0.01$). *NPQ* showed extremely significant relationships with salt concentration ($p < 0.001$) and also exhibited a significant relationship with water–salt interaction stress ($p < 0.05$), but displayed no significant relationship with the groundwater level.

**Table 1.** Correlations of chlorophyll fluorescence parameters to salinity, groundwater level and their interactions.

| Chlorophyll Fluorescence Parameters | Salt Concentration | Groundwater Level | Salt Concentration * Groundwater Level |
|---|---|---|---|
| *Fo* | 2.129(0.065) | 1.612(0.174) | 2.191 ** |
| *Fm* | 2.044(0.076) | 2.891 * | 1.681 * |
| *Fv/Fm* | 14.293 *** | 21.583 *** | 4.928 *** |
| *Fo′* | 57.43 *** | 6.475 *** | 2.079 ** |
| *Fm′* | 68.43 *** | 10.953 *** | 3.593 *** |
| *Fv′/Fm′* | 42.381 *** | 2.077(0.087) | 5.207 *** |
| $\Phi PSII$ | 136.38 *** | 9.231 *** | 11.073 *** |
| *qP* | 42.526 *** | 4.032 ** | 2.623 *** |
| *NPQ* | 67.371 *** | 1.118(0.35) | 1.952 * |
| *ETR* | 60.245 *** | 11.63 *** | 2.387 ** |

Note: The data in the table are Pearson correlation coefficients, *** indicates $p < 0.001$, ** indicates $p < 0.01$, * indicates $p < 0.05$.

The results demonstrated that the chlorophyll fluorescence parameters *Fv/Fm*, *Fm′*, and $\Phi PSII$ of *Suaeda salsa* responded significantly to water and salt stress.

### 3.2. Response Analysis of Suaeda salsa Canopy Reflectance Spectra to Water and Salt Stress

Water and salt stress affected the reflectance of original spectra of the canopy of *Suaeda salsa* (Figure 6). As Figure 1 shows, under different groundwater levels, salt concentrations, and water–salt interactions, the original spectra exhibited a higher raw reflectance within the 500–600 nm, 680–760 nm, 760–920 nm, 1000–1100 nm, 1200–1300 nm, 1370–1400 nm, 1500–1800 nm, and 1800–2350 nm spectral regions. In addition, the original spectra of the canopy of *Suaeda salsa* under the same water level first increased and then decreased with increased salt concentrations.

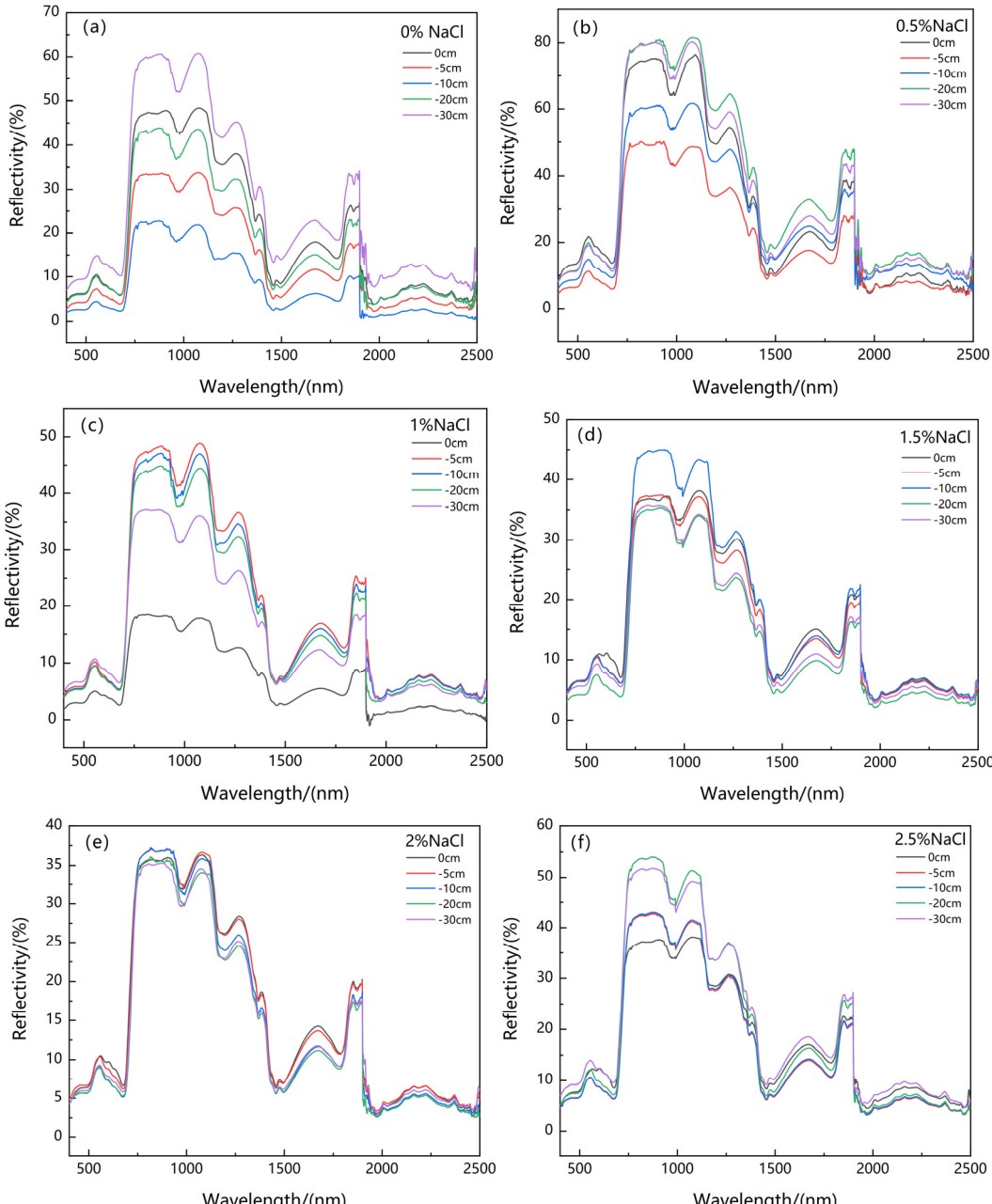

**Figure 6.** The change of reflectance spectrum under five groundwater levels (0 cm, −5 cm, −10 cm, −20 cm, and −30 cm) and six NaCl salt concentrations (0% (**a**), 0.5% (**b**), 1% (**c**), 1.5% (**d**), 2% (**e**), and 2.5% (**f**)).

From Figure 6a, when the 400–2500 nm spectral regions were under conditions of 0% salt concentration, they showed the highest reflectance at a water level of −30 cm, and the lowest reflectance at a water level of −10 cm. The results showed that the deeper groundwater level resulted in the higher reflectance under no salt stress.

From Figure 6b, when the 400–700 nm visible spectral regions were under conditions of 0.5% salt concentration, they showed the highest reflectance at a water level of 0 cm and the lowest reflectance at a water level of −5 cm. Meanwhile, when the 700–2500 nm near infrared and shortwave spectral regions were under conditions of 0.5% salt concentration, they showed the highest reflectance at a water level of −20 cm and the lowest reflectance at a water level of −5 cm.

From Figure 6c, when the 400–700 nm visible spectral regions were under conditions of 1% salt concentration, they showed the highest reflectance at a water level of −30 cm

and the lowest reflectance at a water level of 0 cm. Meanwhile, when the 700–2500 nm near infrared and shortwave spectral regions were under conditions of 1% salt concentration, they showed the highest reflectance at a water level of −5 cm but the lowest reflectance at a water level of 0 cm.

From Figure 6d, when the 400–700 nm visible spectral regions were under conditions of 1.5% salt concentration, they showed the highest reflectance at the water level of 0 cm and the lowest reflectance at a water level of −20 cm. Meanwhile, when the 700–2500 nm near infrared and shortwave spectral regions were under conditions of 1.5% salt concentration, they showed the highest reflectance at a water level of −10 cm and the lowest reflectance at a water level of −20 cm.

From Figure 6e, when it in the 400–2500 nm spectral regions and under the condition of 2% salt concentration showed the same reflectance at different water depths tends.

From Figure 6f, when the 400–700 nm visible spectral regions were under conditions of 2.5% salt concentration, they tended to show the same reflectance at different water depths tends. When the 700–1300 nm near infrared spectral regions were under conditions of 2.5% salt concentration, they showed the highest reflectance at a water level of −20 cm and the lowest reflectance at a water level of 0 cm. Meanwhile, when the 1300–2500 nm shortwave spectral regions were under conditions of 2.5% salt concentration, they showed the highest reflectance at a water level of −30 cm and the lowest reflectance at a water level of −5 cm.

In general, there is little difference in reflection curve in the visible light band from 400 to 700 nm, but the reflection spectrum curve in the near-infrared band from 700 to 750 nm increases significantly, perhaps related to increases in the chlorophyll content per unit area of the plant. Spectral reflectance in the 750–1300 nm near-infrared band was very high and appeared as "infrared high steps" as a result of changes in plants' water absorption and canopy structure. Spectral reflectance decreases in the 1300–2500 nm shortwave infrared range, mainly dependent on the moisture content of vegetation [13].

Analysis of the Response of First-Order Differential Spectrum of *Suaeda salsa* Canopy to Water and Salt Stress

Water and salt stress affected the reflectance of the first-order differential spectra of the canopy of *Suaeda salsa* (Figure 7). As Figure 7 shows, the spectra under different groundwater levels, salt concentrations, and water–salt interactions exhibited a higher raw reflectance within the 680–750 nm, 480–560 nm, 950–1000 nm, 1800–1850 nm, and 1890–1910 nm spectral regions.

*3.3. Correlation Analysis of Suaeda salsa Chlorophyll Fluorescence Parameters and Spectrum Ratio Vegetation Index*

A strong relationship was found between the *Suaeda salsa* chlorophyll fluorescence parameter and the spectrum ratio vegetation index, caused by differences in water salt interaction. To obtain the correlation coefficient matrix, the MatlabR2017b software was used to analyze the correlation between chlorophyll fluorescence parameters and the spectral ratio vegetation index. Figure 8 shows the correlation coefficient and significance level matrix between the chlorophyll fluorescence parameter and the spectrum ratio vegetation index. A higher correlation coefficient showed a gradual yellow color, whereas a lower correlation coefficient showed a gradual blue color. Since no significant correlation exists between *Fo* and *Fm* and the spectral vegetation index, their figures are not listed here. Based on the higher hotspots (as Figure 5 shows), eight single bands (681 nm, 704 nm, 763 nm, 816 nm, 928 nm, 1046 nm, 1897 nm, and 1969 nm) were finally selected to construct the new index, all of which happened to fall within the band range where the spectrum of *Suaeda* canopy was sensitive to water and salt conditions.

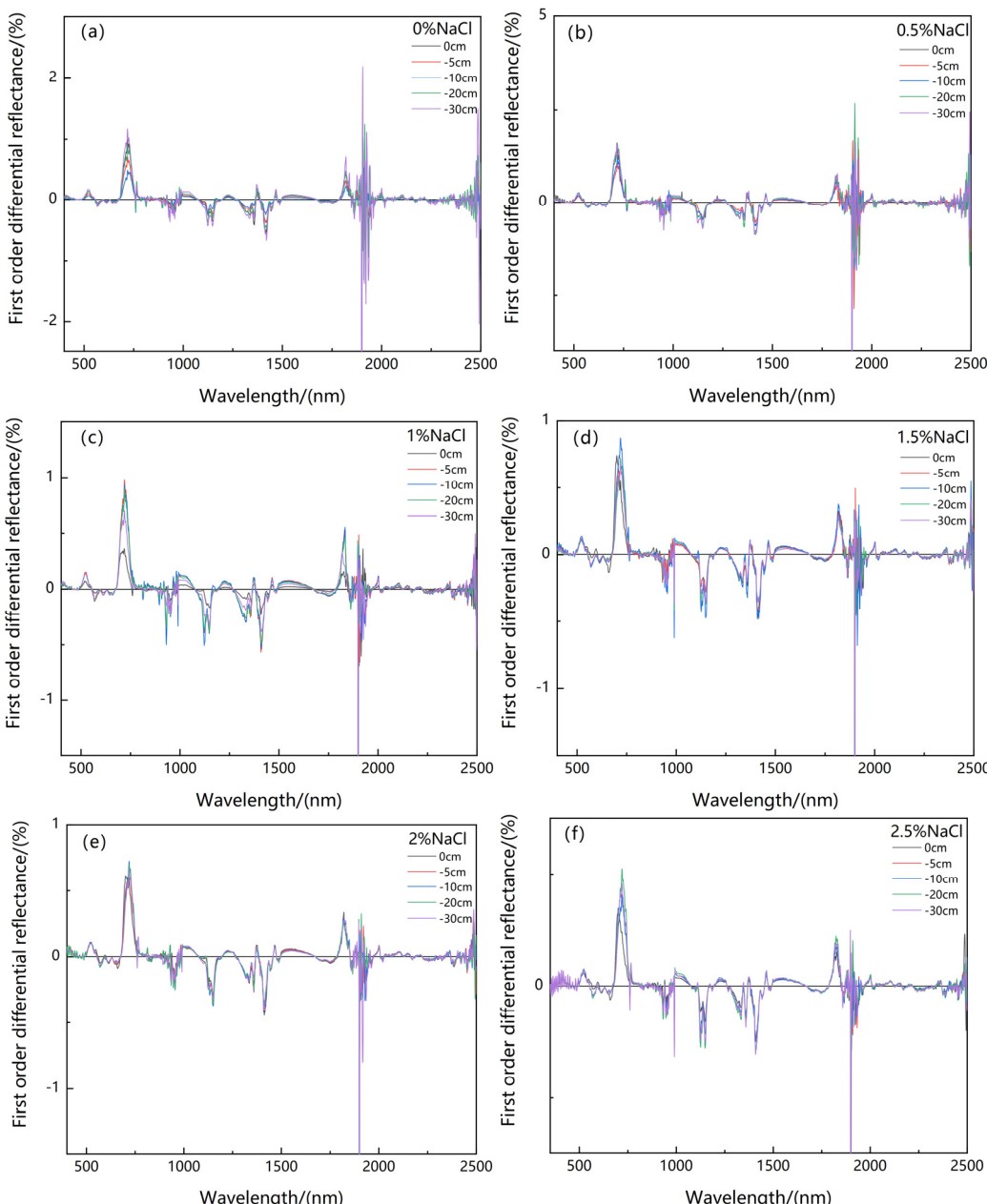

**Figure 7.** The change of the first-order differential reflectance spectrum under five groundwater levels (0 cm, −5 cm, −10 cm, −20 cm, and −30 cm) and six NaCl salt concentrations (0% (**a**), 0.5% (**b**), 1% (**c**), 1.5% (**d**), 2% (**e**), and 2.5% (**f**)).

*3.4. Correlation Analysis of Suaeda salsa Chlorophyll Fluorescence Parameters and First-Order Differential Spectrum Ratio Vegetation Index*

There was a strong relation between the *Suaeda salsa* chlorophyll fluorescence parameter and first-order differential spectral ratio vegetation index, caused by differences in water–salt interaction. To obtain the correlation coefficient matrix, the MatlabR2017b software was used to analyze the correlation between the chlorophyll fluorescence parameter and first-order differential spectral ratio vegetation index. Figure 9 shows the correlation coefficient and significance level matrix between the chlorophyll fluorescence parameter and spectrum ratio vegetation index. A higher correlation coefficient showed a gradual yellow color, whereas the lower correlation coefficient showed a gradual blue color. Similarly, since no significance exists between *Fo* and *Fm*, they are not listed here. Based on the higher hot spots (as Figure 6 shows), ten single bands (486 nm, 668 nm, 690 nm, 725 nm, 948 nm, 954 nm, 980 nm, 1284 nm, 1320 nm, and 1480 nm) were finally selected to construct

the new index, all of which happened to fall within the band range where the spectrum of *Suaeda* canopy was sensitive to water and salt conditions.

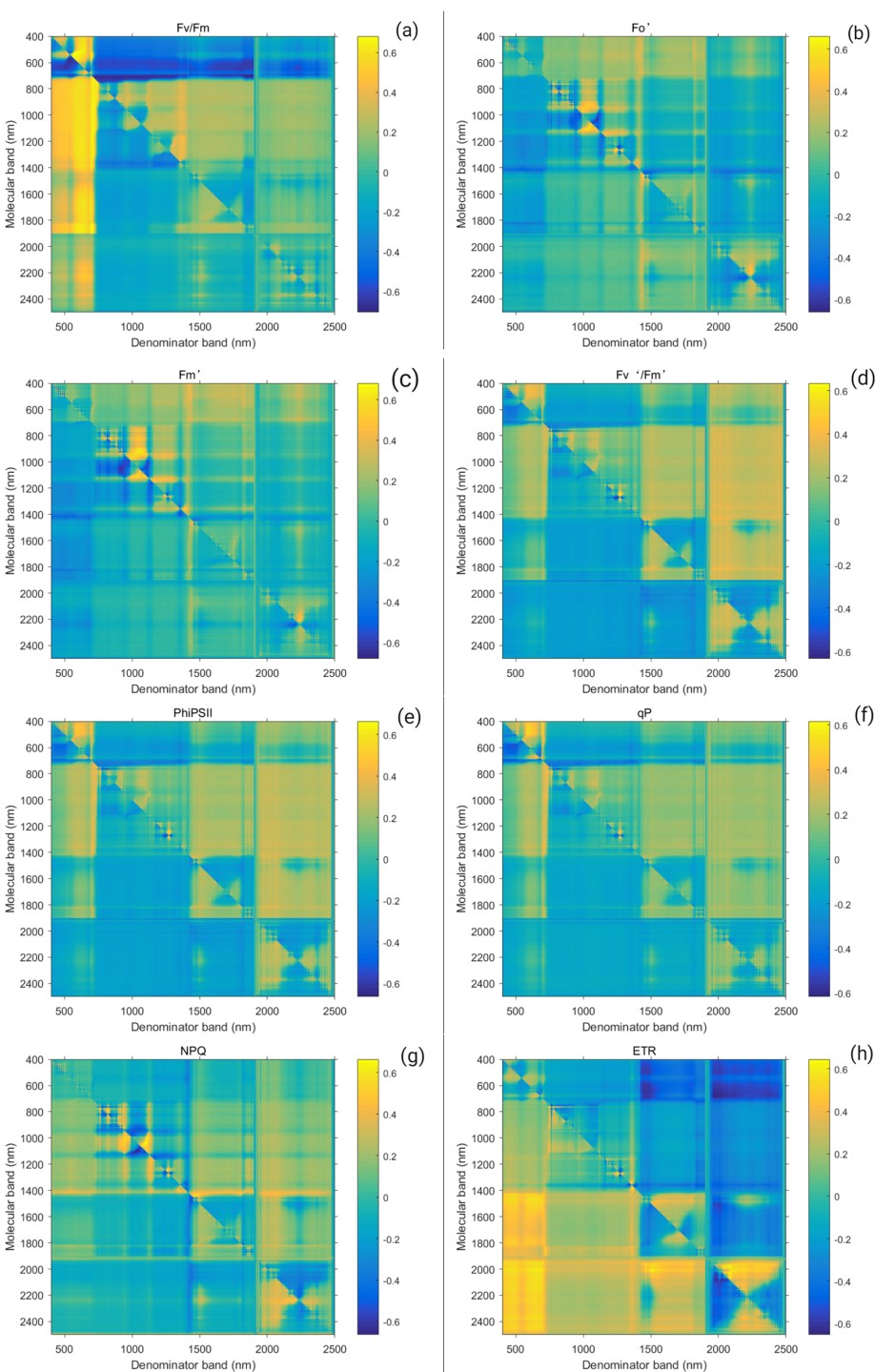

**Figure 8.** Correlation coefficient of chlorophyll fluorescence parameters ($Fv/Fm$ (**a**), $Fo'$ (**b**), $Fm'$ (**c**), $Fv'/Fm'$ (**d**), $\Phi PSII$ (**e**), $qP$ (**f**), $NPQ$ (**g**), $ETR$ (**h**))and spectral ratio coefficient under water and salt conditions.

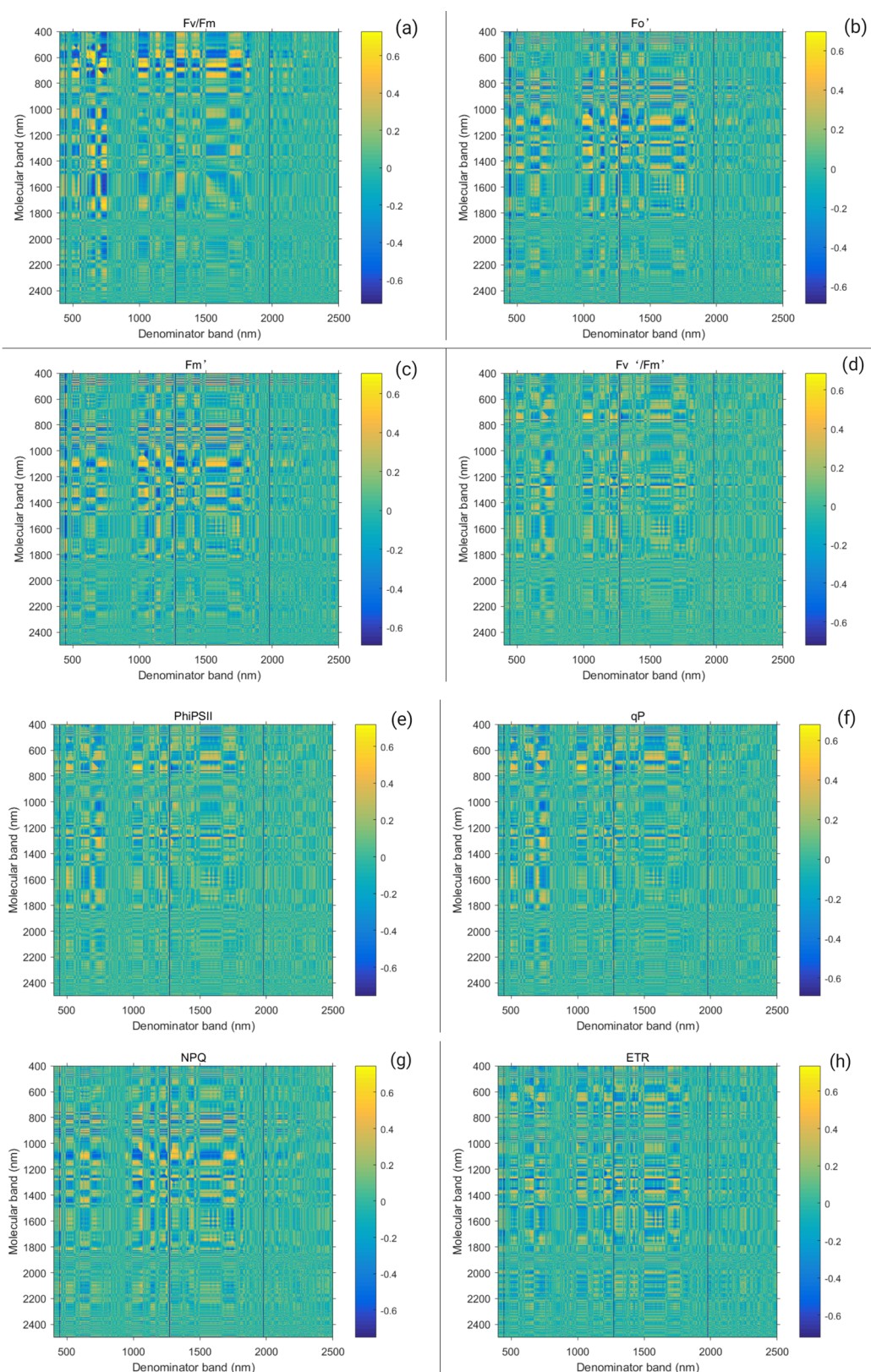

**Figure 9.** Correlation coefficient of chlorophyll fluorescence parameters ($Fv/Fm$ (**a**), $Fo'$ (**b**), $Fm'$ (**c**), $Fv'/Fm'$ (**d**), $\Phi PSII$ (**e**) , $qP$ (**f**), $NPQ$ (**g**), $ETR$ (**h**)) and first-order differential spectrum ratio coefficient under water and salt conditions.

### 3.5. Hyperspectral Index Identification Sensitive to Suaeda salsa Chlorophyll Fluorescence Parameters

Table 2 shows the newly developed and two published vegetation index with references.

**Table 2.** Correlation analysis of hyperspectral vegetation index and chlorophyll fluorescence parameters under water and salt conditions.

| Vegetation Index/Chlorophyll Fluorescence Parameters | $Fv/Fm$ | $Fo'$ | $Fm'$ | $Fv'/Fm'$ | $\Phi PSII$ | $qP$ | $NPQ$ | $ETR$ |
|---|---|---|---|---|---|---|---|---|
| $R_{704}/R_{1897}$ | **−0.706 \*\*** | 0.188(0.076) | 0.142(0.182) | −0.231 \* | −0.296 \*\* | −0.302 \*\* | −0.057(0.596) | −0.412 \*\* |
| $R_{928}/R_{1046}$ | 0.016(0.881) | 0.529 \*\* | **0.643 \*\*** | 0.082(0.441) | 0.084(0.432) | 0.103(0.335) | −0.527 \*\* | −0.150(0.158) |
| $R_{681}/R_{1969}$ | −0.328 \*\* | −0.007(0.948) | 0.112(0.293) | 0.228 \* | 0.189(0.074) | 0.119(0.263) | 0.146(0.170) | **−0.653 \*\*** |
| $D_{690}/D_{1320}$ | **0.716 \*\*** | −0.077(0.472) | −0.020(0.854) | 0.225 \* | 0.325 \*\* | 0.365 \*\* | 0.019(0.862) | 0.325 \*\* |
| $D_{980}/D_{1284}$ | −0.168(0.114) | **0.697 \*\*** | 0.631 \*\* | −0.350 \*\* | −0.357 \*\* | −0.255 \* | −0.589 \*\* | 0.070(0.514) |
| $D_{486}/D_{668}$ | 0.135(0.204) | 0.454 \*\* | **0.705 \*\*** | 0.340 \*\* | 0.356 \*\* | 0.316 \*\* | −0.368 \*\* | −0.407 \*\* |
| $D_{725}/D_{1284}$ | −0.283 \*\* | 0.365 \*\* | 0.085(0.425) | **−0.722 \*\*** | **−0.753 \*\*** | **−0.688 \*\*** | v0.306 \*\* | 0.245 \* |
| $D_{1480}/D_{954}$ | 0.046(0.666) | **−0.617 \*\*** | −0.566 \*\* | 0.185(0.080) | 0.166(0.119) | 0.094(0.381) | **0.745 \*\*** | −0.007(0.945) |
| $D_{948}/D_{724}$ | 0.343 \*\* | −0.053(0.619) | −0.197(0.063) | −0.270 \* | −0.204(0.053) | −0.110(0.302) | 0.014(0.896) | **0.736 \*\*** |
| $(R_{780}-R_{710})/(R_{780}-R_{680})$ | **0.710 \*\*** | −0.070(0.515) | 0.026(0.810) | 0.310 \*\* | 0.386 \*\* | 0.402 \*\* | 0.022(0.837) | 0.252 \* |
| $(R_{850}-R_{710})/(R_{850}-R_{680})$ | **0.708 \*\*** | −0.066(0.536) | 0.024(0.822) | 0.300 \*\* | 0.380 \*\* | 0.401 \*\* | 0.016(0.878) | 0.268 \* |
| $(R_{1897}-R_{704})/(R_{1897}-R_{704})$ | **0.698 \*\*** | -0.216 \* | −0.174(0.101) | 0.228 \* | 0.294 \*\* | 0.294 \*\* | 0.082(0.445) | 0.404 \*\* |
| $(D_{1320}-D_{690})/(D_{1320}+D_{690})$ | 0.080(0.453) | −0.063(0.557) | −0.123(0.247) | −0.070(0.510) | −0.079(0.460) | −0.086(0.422) | 0.054(0.612) | −0.017(0.871) |
| $(D_{1284}-D_{724})/(D_{1284}+D_{724})$ | 0.352 \*\* | −0.250 \* | 0.005(0.959) | **0.666 \*\*** | **0.706 \*\*** | **0.678 \*\*** | 0.178(0.093) | −0.139(0.190) |
| $(D_{1480}-D_{954})/(D_{1480}+D_{954})$ | 0.038(0.723) | **−0.630 \*\*** | −0.575 \*\* | 0.192(0.070) | 0.164(0.122) | 0.083(0.435) | **0.740 \*\*** | 0.000(0.998) |

Note: The data in the table are Pearson correlation coefficients, \*\* indicates $p < 0.01$, \* indicates $p < 0.05$. The higher level of significance in the table has been bolded.

From Table 3, $Fo$ and $Fm$ showed no significant relationships with all vegetation indices, therefore they are not listed in the table. SPSS17.0 software was used to calculate the Person correlation coefficient by bivariate correlation analysis. It can be seen from Table 3 that $R_{704}/R_{1897}$ has the highest correlation coefficient with $Fv/Fm$. $R_{928}/R_{1046}$ has the highest correlation coefficient with $Fm'$. The correlation coefficient between $R_{681}/R_{1969}$ and $ETR$ is the highest. $D_{690}/D_{1320}$ has the highest correlation coefficient with $Fv/Fm$. $D_{980}/D_{1284}$ has the highest correlation coefficient with $Fo'$. $D_{486}/D_{668}$ has the highest correlation coefficient with $Fm'$. $D_{724}/D_{1284}$ has the highest correlation coefficient with $\Phi PSII$. $D_{1480}/D_{954}$ has the highest correlation coefficient with $NPQ$. $D_{948}/D_{724}$ has the highest correlation coefficient with $ETR$. $(R_{780}-R_{710})/(R_{780}-R_{680})$ has the highest correlation coefficient with $Fv/Fm$. $((R_{850}-R_{710})/(R_{850}-R_{680})$ has the highest correlation coefficient with $Fv/Fm$. $(R_{1897}-R_{704})/(R_{1897}-R_{704})$ has the highest correlation coefficient with $Fv/Fm$. $(D_{1320}-D_{690})/(D_{1320}+D_{690})$ showed no significant relationships with these eight chlorophyll fluorescence parameters. $(D_{1284}-D_{724})/(D_{1284}+D_{724})$ has the highest correlation coefficient with $\Phi PSII$. $(D_{1480}-D_{954})/(D_{1480}+D_{954})$ has the highest correlation coefficient with $NPQ$. Our result revealed that the new vegetation index constructed in this study demonstrated stronger relationships with chlorophyll fluorescence parameters. According to the correlation coefficient between vegetation index and chlorophyll fluorescence parameters, $D_{690}/D_{1320}$ could be used for retrieving $Fv/Fm$, $D_{980}/D_{1284}$ could be used for retrieving $Fo'$, $D_{486}/D_{668}$ could be used for retrieving $Fm'$, $D_{724}/D_{1284}$ could be used for retrieving $Fv'/Fm'$, $\Phi PSII$, and $qP$, $D_{1480}/D_{954}$ could be used for retrieving $NPQ$, and $D_{948}/D_{724}$ could be used for retrieving $ETR$.

**Table 3.** Validation of chlorophyll fluorescence parameters $Fv/Fm$ and $\Phi PSII$ model for hyperspectral vegetation retrieval.

| Numerical Comparison/ Correlation Analysis | Pearson Correlation | Significance | *RMSEP* |
|---|---|---|---|
| $D_{690}/D_{1320}$ calculates the value of $Fv/Fm$ (predicted value) $Fv/Fm$ measured value | 0.548 ** | 0.002 | 0.334 |
| $D_{725}/D_{1284}$ calculates the value of $\Phi PSII$ (predicted value) $Fv/Fm$ measured value Measured value of $\Phi PSII$ | 0.779 ** | 0.000 | 0.388 |

Note: The data in the table are Pearson correlation coefficients, ** indicates $p < 0.01$.

### 3.6. Construction of Hyperspectral Recognition Model of Suaeda salsa Chlorophyll Fluorescence Parameters

Figure 10 shows that $F_v/F_m$, $Fo'$, $Fm'$, $NPQ$ and $ETR$ had significant and positive relationships with vegetation index, whereas $\frac{Fv'}{Fm'}$, $\Phi PSII$, and $qP$ showed significant and negative relationships with vegetation index. The relationship between chlorophyll fluorescence parameters and vegetation index with the curvilinear were the best model. We selected the spectral feature indexes $D_{690}/D_{1320}$, $D_{980}/D_{1284}$, $D_{486}/D_{668}$, $D_{725}/D_{1284}$, and $D_{1480}/D_{954}$, which were significantly related to the chlorophyll fluorescence parameters of *Suaeda salsa* to construct an inversion model and calculate its multiple determination coefficient (R$^2$) and root mean square error (*RMSE*). The accuracies of the model from high to low are Figure 10e, Figure 10a, Figure 10d, Figure 10f, Figure 10h, Figure 10g, Figure 10b, and Figure 10c.

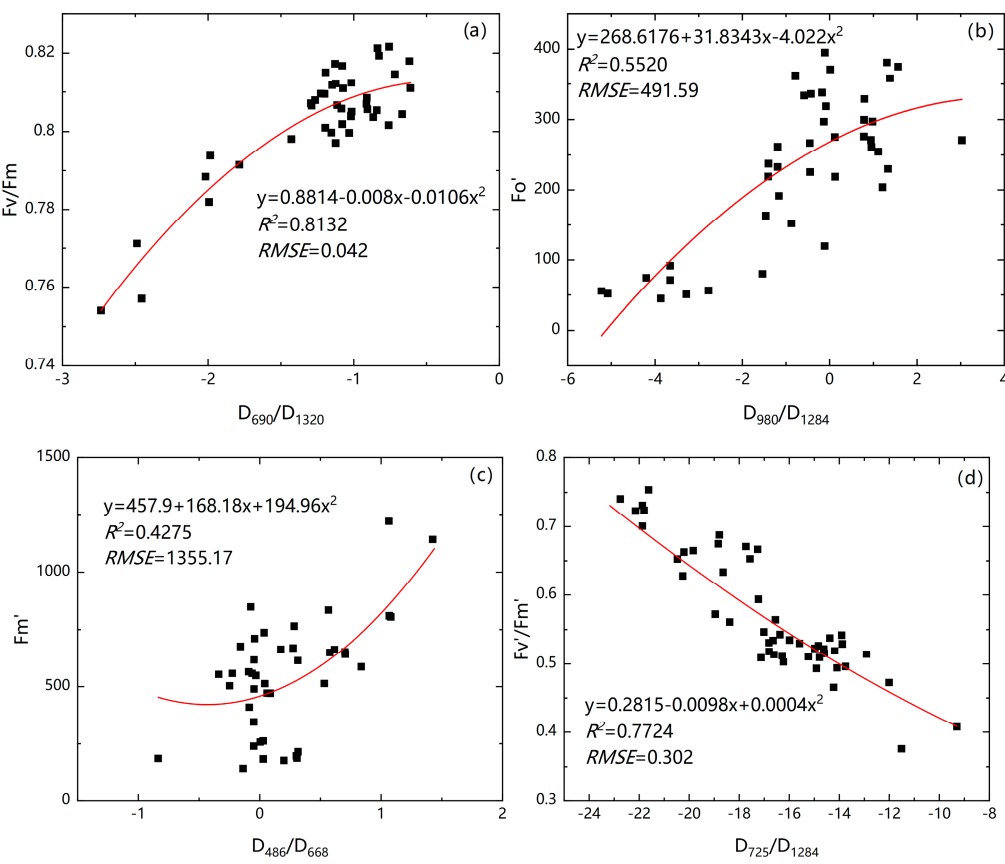

**Figure 10.** *Cont.*

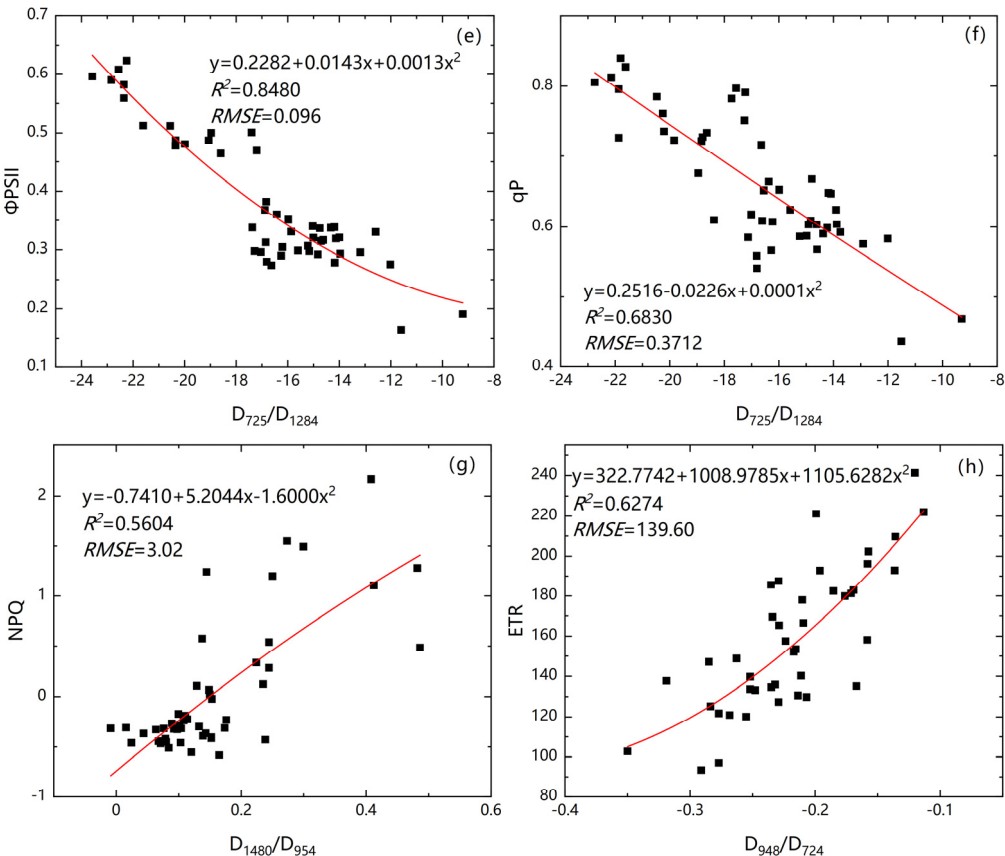

**Figure 10.** Fitting relationship diagram between the vegetation index with the highest correlation and the corresponding chlorophyll fluorescence parameter. ((**a**) fitting equation of $D_{690}/D_{1320}$ and $Fv/Fm$; (**b**) fitting equation of $D_{980}/D_{1284}$ and $Fo'$; (**c**) fitting equation of $D_{486}/D_{668}$ and $Fm'$; (**d**) fitting equation of $D_{725}/D_{1284}$ and $Fv'/Fm'$; (**e**) fitting equation of $D_{725}/D_{1284}$ and $\Phi PSII$; (**f**) fitting equation of $D_{725}/D_{1284}$ and $qP$; (**g**) fitting equation of $D_{1480}/D_{954}$ and $NPQ$; (**h**) fitting equation of $D_{948}/D_{724}$ and $ETR$). The red curve represents the best fit line.

### 3.7. Verification and Evaluation of the Accuracy of the Hyperspectral Recognition Model of Suaeda salsa Chlorophyll Fluorescence Parameters

From Figure 10a,e, $D_{690}/D_{1320}$ and $D_{725}/D_{1284}$, with higher correlation coefficient ($R^2$) and lower *RMSE*, could be used to retrieve the chlorophyll fluorescence parameters $Fv/Fm$ and $\Phi PSII$.

We used the one-variable quadratic polynomial, obtained by the inversion model to receive the predicted value, and then evaluated the accuracy of the model based on the Pearson correlation coefficient between the predicted value and the measured value with the *RMSE*. Table 3 shows that $\Phi PSII$ exhibited a higher correlation between measured and predicted values.

The chlorophyll fluorescence parameters of *Suaeda salsa* predicted by the model were compared with the measured values. As Figure 8 shows, the abscissa was the measured value and the ordinate was the predicted value. It can be seen that the points were uniformly distributed near the straight line y = x, whereas $D_{725}/D_{1284}$ retrieved chlorophyll fluorescence parameter $\Phi PSII$'s prediction model slope was 1.076 (Figure 11b) and $D_{690}/D_{1320}$ retrieved chlorophyll fluorescence parameter $Fv/Fm$'s prediction model slope was 0.606 (Figure 11a).

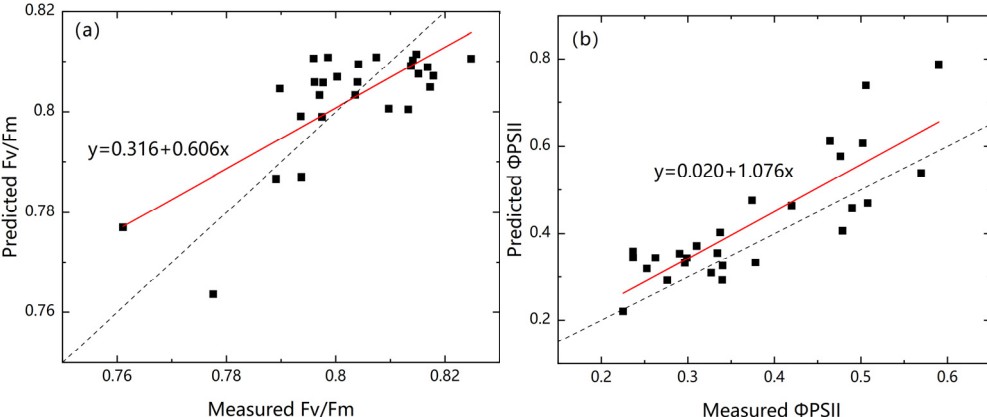

**Figure 11.** Predictive models of $Fv/Fm$ (**a**) and $\Phi PSII$ (**b**). The dotted line represents the prediction model in an ideal state, and the red line represents the real prediction model.

In general, the prediction model of $D_{725}/D_{1284}$ inverted the chlorophyll fluorescence parameter $\Phi PSII$ has a higher accuracy.

## 4. Discussion

Moisture and salinity were the key factors restricting the growth of coastal wetland plants [28]. Water stress can affect the chlorophyll content of plant leaves, whereas the photosynthetic capacity of plant leaves was affected by the level of chlorophyll content [29]. Salt stress damaged the photosynthetic organs of plant chloroplasts and destroyed the reaction center of the photosystem, leading to a decline in photosynthetic capacity and indirectly affecting plant growth [30]. In addition, the greater the salinity, the longer and more obvious the action time was [9]. The chloroplast photosystem II (PSII) in the leaves was affected first under water and salt stress [31].

Consistent with the results of our study, which found that the chlorophyll fluorescence parameters $Fv/Fm$, $Fm'$, and $\Phi PSII$ of *Suaeda salsa* showed significant relationships with vegetation index water and salt (Table 1), Zhang et al. [26] found that $Fm$, $\frac{Fv}{Fm}$, and $qP$ decreased significantly under salt stress. Yin et al. [32] found that the chlorophyll content of *Suaeda salsa* leaves significantly decreased in a severely salty environment compared with moderately and lightly salted environments, whereas the light energy utilization rate gradually decreased. In medium and light salinity environments, $Fo$ and $Fm$ were highly reduced. *Suaeda salsa* inhibited photosynthesis under water and salt stress. Furthermore, PSII activity was lower under increased salinity, with photosynthetic capacity damaged to a certain extent.

Water and salt stress has a significant impact on the spectral reflectance of *Suaeda salsa*'s canopy spectrum and first-order differential spectrum in the bands 680–750 nm, 480–560 nm, 950–1000 nm, 1800–1850 nm, and 1890–1910 nm (Figure 6). Our result is similar to that of Wu [21], who found that the blue valley peak band of the visible light range of green bamboo leaves in the spectral range 420–500 nm, the reflection peak of the green light band in the visible light range 550 nm, the visible light range of red light in the spectral range 600–680 nm, the red edge area in the spectral range 680–740 nm, and near-infrared band in the spectral range 780–1000 nm all have typical reflectance spectral characteristics under salt stress. Salah El-Hendawy et al. [33] obtained a new vegetation index from the information of photosynthetic pigments and fluorescence emission bands, as well as visible light (531 nm, 550 nm, 570 nm, 630 nm, 670 nm, and 690 nm), near infrared (800 nm, 870 nm, 970 nm, 1050 nm, 1100 nm, and 1250 nm) and short-wave infrared (1450 nm, 1650 nm, and 2250 nm) photosynthesis. The new vegetation index constructed in this article comes from visible light (486 nm, 668 nm, 681 nm, and 690 nm), near infrared (704 nm, 725 nm, 763 nm, 816 nm, 928 nm, 948 nm, 954 nm, 1046 nm, and 1284 nm), and

short-wave infrared (1320 nm, 1480 nm, 1897 nm, and 1969 nm) were also relatively close to the sensitive bands (Figures 8 and 9), so the research results are essentially the same.

The new vegetation index explored in this article was mainly in the form of a/b and (a − b)/(a + b) (Table 3). Zhang et al. [31] selected two bands of 680 nm and 935 nm according to the most sensitive hyperspectral bands of *Suaeda* chlorophyll fluorescence parameters and ultimately proved that $(R_{680} - R_{935})/(R_{680} + R_{935})$ and $R_{680}/R_{935}$ have a higher determination coefficient ($R^2$) and lower root mean square error ($RMSE$). Yoshio et al. [34] explored models for predicting the chlorophyll content of the canopy based on the spectral data set of six planting types (rice, wheat, corn, soybean, sugar beet, and natural grass) and found that that the simple ratio spectrum model was more accurate and applicable than that of the multivariate regression model. Accordingly, we chose $D_{690}/D_{1320}$ and $D_{725}/D_{1284}$, which have the highest correlation coefficients, to retrieve the chlorophyll fluorescence parameters $Fv/Fm$ and $\Phi PSII$ (Table 3, Figure 11). Zarco-Tejada et al. [35] proved that the double-peak feature on canopy derivative reflectance is entirely due to the chlorophyll fluorescence effect and the importance of the derivative method in fluorescence detection in vegetation.

Murchie and Lawson [36] found that $\Phi PSII$ decreased under salt stress due to a reduction in stomatal conductance, and Zhang [37] discovered that $Fv/Fm$ refers to the maximum photochemical efficiency of PSII, reflecting the original light energy in the PSII reaction center. The fluorescence parameter $Fv/Fm$ changes greatly under environmental stress conditions, indicating that $Fv/Fm$ is an important parameter of the photochemical reaction status.

This article finally constructed new vegetation indices based on the reflectance of the first-order differential of the *Suaeda* canopy spectrum to retrieve the two chlorophyll fluorescence parameters $Fv/Fm$ and $\Phi PSII$.

In addition to moisture and salinity, the interaction of plants and animals in wetlands can result in stresses in plant. One example is the fouling of barnacles on mangrove plant bark and leaves resulted in reduction of gaseous exchanges. Plants subsequently demonstrate reduced chlorophyll levels and develop more stomata to enhance gaseous exchange [38]. Similarly, Pomacea snails also create severe impacts on wetland plants [39,40] in Asian regions. The present remote sensing technique (chlorophyll fluorescence parameters) can also be applied to monitor plant health in cases of pest stress, as this is an important issue in conservation and management of coastal wetlands.

## 5. Conclusions

Taking *Suaeda salsa* in coastal wetlands as the research object, we set up five groundwater levels (0 cm, −5 cm, −10 cm, −20 cm, and −30 cm) and six NaCl salt concentrations (0%, 0.5%, 1%, 1.5%, 2%, and 2.5%) to carry out an independent test of *Suaeda salsa* potted plants, measuring the canopy reflectance spectrum and chlorophyll fluorescence parameters of *Suaeda salsa*. We used a polynomial regression method to carry out hyperspectral identification of *Suaeda salsa* chlorophyll fluorescence parameters under water and salt stress, arriving following conclusions:

(1)   The chlorophyll fluorescence parameters $Fv/Fm$, $Fm'$, and $\Phi PSII$ of *Suaeda salsa* showed significant relationships with the vegetation index under water and salt. The spectra under different groundwater levels, salt concentrations, and water–salt interactions exhibited a higher raw reflectance within the 500–600 nm, 680–760 nm, 760–920 nm, 1000–1100 nm, 1200–1300 nm, 1370–1400 nm, 1500–1800 nm, and 1800–2350 nm spectral regions. However, when we measured the reflectance of the canopy spectrum, we found that the reflectance of the canopy spectrum may also be related to light conditions, soil types, and crop varieties [41]. More studies are needed to verify our findings [42].

(2)   We constructed thirteen new vegetation indices. In addition, we discovered that the model using hyperspectral vegetation index $D_{690}/D_{1320}$ (the simple ratio of the derivative) to retrieve the *Suaeda* chlorophyll fluorescence parameter $Fv/Fm$ was the

most accurate, with a multiple determination coefficient $R^2$ of 0.813 and an *RMSE* of 0.042. $D_{725}/D_{1284}$ (the simple ratio of the derivative) retrieved the *Suaeda* chlorophyll fluorescence parameter $\Phi PSII$ model with the highest accuracy, with a multiple determination coefficient $R^2$ of 0.848 and an *RMSE* of 0.096. However, it remains to be seen whether the newly proposed vegetation index can be applied to the model of UAV hyperspectral remote sensing imagery being constructed to estimate the chlorophyll fluorescence parameters of *Suaeda salsa* over a large area under water and salt conditions. Thus, subsequent verification is needed.

**Author Contributions:** Conceptualization, Y.Z., W.Z. and X.L.; methodology, W.Z.; software, W.Z.; validation, W.Z., X.L. and S.L.; formal analysis, W.Z.; investigation, W.Z.; resources, W.Z.; data curation, W.Z.; writing—original draft preparation, W.Z.; writing—review and editing, X.L. and Y.Z.; visualization, X.L.; supervision, Y.Z. and Y.L.; project administration, X.L.; funding acquisition, X.L. All authors have read and agreed to the published version of the manuscript.

**Funding:** This study was financially supported by National Natural Science Foundation of China (U1901215 & 41506106), the Jiangsu Natural Science Foundation of China (BK20171262), Lianyungang City "Haiyan Plan" project "Evaluation of Coastal Wetland Ecological Restoration Effect" in 2019, Lianyungang City Huaguoshan Talents-Vice Master Plan Project for Science and Technology, Jiangsu University Superior Discipline Construction Project Funding Project (PAPD)), Jiangsu Province Marine Technology First-Class Professional Construction Project, 2020 Jiangsu Province Graduate Practical Innovation Project "Analysis of Spatial and Temporal Distribution Characteristics and Driving Factors of Vegetation in the Yellow River Delta"(SJCX20_1242), and 2020 Jiangsu Province Graduate Practical Innovation Project "Estimation of Porphyra Biomass in Jiangsu Coastal Area by Remote Sensing"(SJCX20_1247).

**Institutional Review Board Statement:** Not applicable.

**Informed Consent Statement:** Not applicable.

**Data Availability Statement:** All data comes from our actual measurement, and there is no conflict of interest with others.

**Acknowledgments:** We thank Yali Lin, Seng Zhang, and Yexin Zhang's support for data collection and measurements. We also thank two reviewers for their technical suggestions and critical comments. The research was partially supported by the National Natural Science Foundation of China (U1901215 & 41506106), the Jiangsu Natural Science Foundation of China (BK20171262), the 2019 Lianyungang City "Haiyan Plan" project "Evaluation of Coastal Wetland Ecological Restoration Effect", Lianyungang City Huaguoshan Talents-Vice Master Plan for Science and Technology Project, Jiangsu University Superior Discipline Construction Project Funding Project (PAPD), Jiangsu Marine Technology First-Class Professional Construction Project, 2020 Jiangsu Graduate Practice Innovation Project "Analysis of Spatial and Temporal Distribution Characteristics and Driving Factors of Vegetation in the Yellow River Delta" (SJCX20_1242) and 2020 Jiangsu Province Graduate Practical Innovation Project "Estimation of Porphyra Biomass in Jiangsu Coastal Area by Remote Sensing"(SJCX20_1247).

**Conflicts of Interest:** The author declares that this article has no conflict of interest.

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
