# Peer review of "Hyperspectral Identification of Chlorophyll Fluorescence Parameters of Suaeda salsa in Coastal Wetlands"

_remotesensing, doi:10.3390/rs13112066_

Round 1
Reviewer 1 Report
1) The introduction should be elaborated with more general information. At present it stated as “Coastal wetlands are located at the junction of land-sea ecosystems [1]. The dominant halophyte Suaeda can absorb the salt and heavy metals in the soil, increase soil fertility and improve the ecological environment of the coastal wetlands [2,3].”
It should be revised as something like “Coastal wetlands are located at the junction of land-sea ecosystems [1]. As a result, fluctuations in salinity is one of the key features in wetlands, animals and plants under under prolonged stress in variation in salinities (Ball, 1998; Theuerkauff et al. 2018).”
Please cite:
Ball, M. (1998). Mangrove Species Richness in Relation to Salinity and Waterlogging: A Case Study Along the Adelaide River Floodplain, Northern Australia. Global Ecology and Biogeography Letters, 7(1), 73-82. doi:10.2307/2997699
Theuerkauff D, Rivera-Ingraham GA, Roques JAC, Azzopardi L, Bertini M, Lejeune M, Farcy E, Lignot J, Sucré E. 2018. Salinity variation in a mangrove ecosystem: a physiological investigation to assess potential consequences of salinity disturbances on mangrove crabs. Zool Stud 57:36. doi:10.6620/ZS.2018.57-36.
2) The second sentence it jumps suddenly “The dominant halophyte Suaeda can absorb the salt and heavy metals in the soil, increase soil fertility and improve the ecological environment of the coastal wetlands [2,3].”
“Before Suada appear in text, you should state where this plant is dominate. For example, “In the wetlands of the temperate Yellow Sea, the salt marsh plant Suaeda is common.” Then go on to the sentence with Suaeda.
3) The last paragraph of the introduction should end up with a hypothesis. Suaeda chlorophyll fluorescence parameters will change under different salinity conditions.
4) 2.1 should be study sites, not materials. In this section, the author should state the salinity ranges of those ground water or surface water measured in the site. Without this background information, it is difficult to setup your rationale for the salinity you used in the experiments. What is the tide ranges of the site? Will those plant covered by water in the high tides? All these information should be mentioned in details in the study site decription. What is the temperature ranges in the whole year? Summer mean air temperature and winter mean air temperature, all these should be mentioned. Any variation in the salinity variation in summer and winter? I would recommend to add a photo of this plant in the same figure of the map, so people can know this plant’s appearance as this journal is not a plant specialist journal. Also, how to apply the spectroradiometer on the plant should be shown in a photo as well. This is a remote sensing journal, it is good to show the technique.
5) 3.6. what is the meaning of the word “Phyllophyll” Is it Chlorophyll?
6) Result and experiment design is fine.
7) In the last paragraph of the discussion, I would suggest ending in a further direction. In the discussion, the author mentioned moisture and salinity is important factor for plant growth in wetlands. However, in wetlands, there are often animal-plant interaction and also affect plant growth and chlorophyll level. Barnacle are common foulers on the surface of mangrove plants. When it is attached on barks and leaves, the gaseous exchange of plant is reduced and this result in plant to develop more stomata and had an impact of reduction in chlorophyll (Lee et al. 2008). The technique introduced here can apply to monitor plant health in wetland management: I would suggest after the last sentence of discussion, add a few sentence like:
Besides moisture and salinity, interaction of plant and animals in wetlands can result in stresses in plant. One of example is the fouling of barnacles on mangrove plant bark and leaves resulted in reduction of gaseous exchanges. Plant subsequently reduced in chlorophyll level and develop more stomata to enhance gaseous exchange (Li et al. 2008). Similarily, Pomacea snails also create severe impacts on wetland plants (Yang and Yu, 2019; Dumidae et al. 2021) in Asian region. The present remote sensing technique (chlorophyll fluorescence parameters) can also be applied to monitor plant health for plants which suffered from pest stress in wetlands, as this is an important issue in conservation and management of coastal wetlands.
Please cite:
Li, S. W., B. K. K. Chan and N. T. Y. Tam (2008). Effect of barnacle fouling on leaf stomata density and chlorophyll concentration of Kandelia obovata, a dominant mangrove species in Hong Kong and Taiwan. Hydrobiologia, 618: 199-203
Yang QQ, Yu XP. 2019. A new species of apple snail in the genus Pomacea (Gastropoda: Caenogastropoda: Ampullariidae). Zool Stud
58:13. doi:10.6620/ZS.2019.58-13.
Dumidae A., Janthu, P., Subkrasae, C., Polsee, R., Mangkit, B., Thanwisai, A., Vitta, P. (2021). Population Genetics Analysis of a Pomacea Snail (Gastropoda: Ampullariidae) in Thailand and its Low Infection by Angiostrongylus cantonensis. Zoological Studies 60: 31. doi:10.6620/ZS.2021.60-31
Author Response
Dear Reviewer:
Thank you very much for your comments and suggestions.
We have revised and improved as suggested.
Yours sincerely,
Yuanzhi

Reviewer 2 Report
See attached file.

Author Response
Dear Reviewer,
Thank you for your suggestions and comments.
We have revised and improved as suggested.
Best regards,
Yuanzhi
